# Zinc Ionophore Pyrithione Mimics CD28 Costimulatory Signal in CD3 Activated T Cells

**DOI:** 10.3390/ijms25084302

**Published:** 2024-04-12

**Authors:** Jana Jakobs, Lothar Rink

**Affiliations:** Institute of Immunology, Faculty of Medicine, RWTH Aachen University, Pauwelsstraße 30, 52074 Aachen, Germany; jjakobs@ukaachen.de

**Keywords:** zinc, T helper cell 1, signaling, differentiation

## Abstract

Zinc is an essential trace element that plays a crucial role in T cell immunity. During T cell activation, zinc is not only structurally important, but zinc signals can also act as a second messenger. This research investigates zinc signals in T cell activation and their function in T helper cell 1 differentiation. For this purpose, peripheral blood mononuclear cells were activated via the T cell receptor-CD3 complex, and via CD28 as a costimulatory signal. Fast and long-term changes in intracellular zinc and calcium were monitored by flow cytometry. Further, interferon (IFN)-γ was analyzed to investigate the differentiation into T helper 1 cells. We show that fast zinc fluxes are induced via CD3. Also, the intracellular zinc concentration dramatically increases 72 h after anti-CD3 and anti-CD28 stimulation, which goes along with the high release of IFN-γ. Interestingly, we found that zinc signals can function as a costimulatory signal for T helper cell 1 differentiation when T cells are activated only via CD3. These results demonstrate the importance of zinc signaling alongside calcium signaling in T cell differentiation.

## 1. Introduction

T cells are initially activated by binding to a specific antigen-presenting cell. This activates various signaling pathways, most importantly of the T cell receptor (TCR)-CD3 complex, but also costimulatory signals such as CD28 [1,2]. In TCR signaling, calcium signals occur from intracellular and extracellular sources. The phospholipase Cγ1 leads to a calcium release from the endoplasmic reticulum which, in turn, opens calcium release-activated calcium (CRAC) channels in the plasma membrane [3,4]. This leads to the activation of the calcineurin-NFAT pathway which regulates gene expression for T cell development [5].

In addition to calcium, zinc is also important in T cell activation. Zinc is an essential trace element and is important for immune regulation in, [6] for example, antiviral immunity [7], autoimmune diseases [8], and tumor defense mechanisms [9]. Zinc is present in the serum at mean concentrations of 84.9 and 80.6 µg/dL in males and females, respectively [10]. It has, on the one hand, structural and catalytical functions [11,12], while, on the other hand, it acts as an ionic signaling molecule and thus influences signaling cascades [13,14]. TCR signaling is affected by zinc at various levels; for example, the activity of lymphocyte-specific protein tyrosine kinase (Lck) [15,16] and protein kinase C [17] is increased by zinc ions, whereas calcineurin is inhibited by zinc [18,19,20].

Therefore, intracellular zinc homeostasis is highly regulated. There are 14 zinc importers (Zip) which increase intracellular zinc [21,22], while 10 zinc transporters (ZnT) decrease intracellular zinc levels by either exporting or redistributing zinc into intracellular compartments [23]. Further, intracellular zinc is buffered by zinc-binding proteins such as metallothionein [24,25] and the S100 proteins calprotectin and calgranulin [26]. These proteins not only maintain zinc homeostasis but can also release zinc to act as a second messenger [24,27].

Zinc signals can feature different kinetics. Fast zinc signals, also known as zinc flux, occur within seconds of stimulation and act as a second messenger [13,14]. Slower zinc signals, known as zinc waves, occur a few minutes after receptor stimulation and depend on calcium signals [28]. Beyond these fast signals, there are also homeostatic changes in zinc levels following stimulation by regulating zinc transporter expression [14,29].

In T cells, a zinc flux was seen after stimulation with phorbol esters, which release zinc via the protein kinase C [30,31]. Furthermore, the stimulation of the T cell receptor via superantigen-loaded dendritic cells showed a zinc signal in the subsynaptic compartment depending on extracellular zinc and the zinc transporter Zip6 [13]. The upregulation of Zip6 was further highlighted in the following studies [32]. Additionally, Zip8 was increased in T cells stimulated with anti-CD3/CD28/CD2, which was important for interferon (IFN)-γ expression [33], a cytokine produced by T helper (Th) 1 cells [34].

In this study, we further elucidated zinc signals in T cell activation and distinguished zinc flux and homeostatic zinc signals induced either via CD3 or the costimulatory signal CD28. We found that zinc signals can mimic the effects of CD28 signaling on IFN-γ expression.

## 2. Results

### 2.1. Fast Zinc Signals Are Induced via CD3 Signaling

T cells are activated via binding to the TCR-CD3 complex and via coreceptors, such as CD28. With CD3 stimulation, intracellular calcium signals occur; additionally, Yu et al. showed zinc signals in T cells after contact with dendritic cells [13]. Here, we investigated if CD3 and CD28 signaling can also individually induce zinc signals. For this, PBMCs were stimulated with anti-CD28 and anti-CD3 antibodies. Intracellular calcium and zinc signals were monitored by flow cytometry 10 min after stimulation. For calcium signals, we used thapsigargin as positive control, which inhibits the sarco-endoplasmic reticulum calcium ATPase and thus increases cytosolic calcium [35]. For zinc signals, the zinc ionophore pyrithione was used as a positive control [36]. As expected, CD3 and CD3 + CD28 but not CD28 stimulation significantly increased intracellular calcium concentrations (Figure 1a). Investigating zinc, we found that CD3 but not CD28 induces an increase in intracellular free/labile zinc. However, for CD3 + CD28, a significant zinc signal could not be seen.

Since calcium signals are described as occurring within seconds [37], we wondered if zinc signals via CD3 stimulation are also as quickly induced. For this, PBMCs were stained with FluoZin-3 and Fluo-4, respectively, and a baseline signal was measured for 55 s with flow cytometry. Subsequently, cells were stimulated with anti-CD3 beads and fast signals were immediately measured for at least 250 s. Analyzing flow cytometric kinetic data, we were able to differentiate two populations: (1) lymphocytes bound to anti-CD3 beads (CD3_B_^+^) and (2) lymphocytes not bound to anti-CD3 beads (CD3_B_^−^). Anti-CD3 beads increased the side scatter in flow cytometry measurements. Thus, after stimulation with anti-CD3 beads, two additional populations emerged in the forward versus side scatter plot when lymphocytes interacted with anti-CD3 beads (Figure 2a,b). A first gate was chosen for total lymphocytes, and, as a control, PBMCs were stimulated with uncoated beads (Control_B_), and no additional populations were seen (Figure 2c). Within the total lymphocytes, lymphocytes bound to anti-CD3 beads were further distinguished due to a strong autofluorescence over 670 nm (Figure 2d). The kinetic of zinc and calcium signals of these two populations was monitored (Figure 2e,f), and the mean fluorescent signal 200–250 s after anti-CD3 bead stimulation was compared. Not only was a fast calcium signal induced by CD3 stimulation (Figure 2g), but intracellular zinc levels were also significantly increased (Figure 2h). To investigate how the kinetic signals change over a longer time period, we also monitored signals for up to 600 s (10 min) in some experiments, finding that signals were stable over this period (Figure 2i,j).

### 2.2. CD3 and CD28 Signaling Induce a Homeostatic Zinc Signal

In addition to short-term changes, we investigated homeostatic changes in intracellular zinc and calcium levels 24–72 h after CD28 and CD3 stimulation, respectively. We found that the homeostatic calcium level was not significantly changed after CD28, CD3 and CD3 + CD28 stimulation (Figure 3a). However, CD3 and CD3 + CD28 stimulation decreased intracellular free calcium. In contrast, intracellular zinc levels increased with time. CD3 and CD28 individually increased intracellular zinc, whereas the highest zinc concentration was induced with CD3 + CD28 72 h after stimulation (Figure 3b). It seems that CD28 causes a low homeostatic zinc signal, while CD3 induced both a fast zinc flux and a homeostatic zinc signal.

### 2.3. Zinc Mimics CD28 Signaling Relevant for IFN-γ Release

It was shown before that zinc is important for IFN-γ expression [33], which is produced by Th1 cells [34]. For example, in zinc deficiency, the IFN-γ production is decreased [38,39,40]; therefore, we investigated whether the additional increase in zinc after CD3 + CD28 stimulation, compared to CD3 stimulation alone, would also enhance IFN-γ expression. For this, PBMCs were stimulated with anti-CD3 and/or anti-CD28 for 72 h. The combination of CD3 + CD28 resulted in significantly more IFN-γ expression than anti-CD3 stimulation alone (Figure 4a). To investigate if the increase in zinc is relevant for this effect, CD28 was replaced with the zinc ionophore pyrithione. We found that pyrithione can mimic the effect of the CD28 costimulatory signal in CD3 activated T cells regarding IFN-γ expression (Figure 4a). We also investigated IL-10, which is expressed by Th2 cell differentiation, but can also be expressed by other cell types [41], as well as IL-17, which is produced by Th17 cells [42]. We found that pyrithione does not have the same effect as anti-CD28 on IL-10 and IL-17 expression (Figure 4b,c). Pyrithione had even the opposite effect by significantly downregulating CD3-induced IL-17 expression (Figure 4c).

## 3. Discussion

In this study, we further elucidated zinc signals in T cell activation and distinguished zinc flux and homeostatic zinc signals induced either via the CD3-TCR complex or the costimulatory signal CD28. We found that the stimulation of CD3 induces not only fast calcium signals but also fast zinc signals. We measured zinc and calcium signals 200–250 s and 10 min after stimulation via flow cytometry in the whole cells. In our experiments, CD3 and CD3 + CD28 increased intracellular free calcium levels 2- and 7-fold, respectively, whereas zinc was increased from mean 0.063 nM to 0.071 nM 10 min after CD3 stimulation. For zinc signals, Yu et al. showed an influx 1 min after T cell activation in the subsynaptic compartment via dendritic cells [13]. In this study, we distinguished that the fast zinc flux is specifically induced by TCR signaling and not by the costimulatory signal CD28. This fast zinc signal might depend on the zinc transporter Zip6 [13]. CD3 not only caused a fast zinc flux, but further caused a homeostatic increase in intracellular zinc 72 h after stimulation. The homeostatic increase in intracellular zinc could be mediated by upregulation of the zinc transporters Zip1, Zip4, Zip6, Zip9 [32], Zip3, Zip8 and Zip14 [33]. Lee et al. also described an increase in intracellular zinc over time for up to 48 h, but which decreased afterwards with a parallel upregulation of zinc-binding metallothionein [43].

Additionally, we investigated the effects of CD28 costimulation. CD28 is necessary for full T cell activation, and TCR stimulation alone can cause a state of unresponsiveness [44,45]. It has been noted that CD28 enhances TCR signaling; however, it also has single signaling effects [46,47]. In our experiments, we found that CD28 signaling alone slightly but significantly increased the intracellular zinc concentration in 72 h-activated cells. In addition, CD28 increased the homeostatic zinc signal of CD3 several fold. As CD3 and CD28 individually increase the homeostatic zinc level, they may regulate the expression of different zinc transporters that add up when both signaling pathways are stimulated simultaneously. Future studies should therefore investigate whether CD28 individually increases zinc transporter expression and whether zinc transporter profiles differ between CD3- and CD3 + CD28-activated T cells. Together, fully activated T cells accumulate zinc but tightly regulate intracellular free calcium concentrations.

The high intracellular zinc concentration in activated T cells further influences signaling cascades of, for example, cytokine receptors, and is therefore also subsequently important for T cell differentiation. For example, zinc is involved in Th1 differentiation. Elderly people with zinc deficiencies show reduced IFN-γ production after T cell activation [48] and studies have shown that zinc influences IFN-γ mainly on a post-transcriptional level [40]. Our results show that the IFN-γ production in CD3 + CD28 activated cells was significantly higher than in CD3 activated cells, which was shown before [49]. Interestingly, we show that the additional stimulus of CD28 can be mimicked by the zinc ionophore pyrithione. This highlights the mechanism of CD28 increasing homeostatic zinc and thereby regulating signaling pathways. CD28 was additionally described to promote T helper cell 2 differentiation [50] and to be important for regulatory T cells [47,51]. However, pyrithione did not have the same effect as CD28 on IL-10 and IL-17 expression. This emphasizes the importance of zinc signals in Th1 differentiation but not in Th2 and Th17 differentiation. This is consistent with the observed upregulation of Th17 cells in zinc-deficient PBMCs [52] and downregulation of Th17 cells with zinc supplementation in murine experimental autoimmune encephalomyelitis [53]. However, conclusions about T cell subsets due to cytokine expressions are limited because the experiments were performed in PBMC, which is a mixed population.

In summary, this study has further elucidated the occurrence and kinetics of zinc signaling during T cell activation. We have shown that CD3 and CD28 signaling have different effects on zinc homeostasis.

## 4. Materials and Methods

### 4.1. Isolation of Peripheral Blood Mononuclear Cells (PBMC)

After informed consent was obtained, human venous blood was collected from healthy volunteer donors and anticoagulated with sodium heparin (B. Braun, Melsungen, Germany). PBMCs were isolated from whole blood, as described previously [54], and cultivated in RPMI-1640 medium (Sigma-Aldrich, Darmstadt, Germany) supplemented with 10% heat-inactivated fetal calf serum (FCS) “Low Endotoxin” (Bio and Sell, Feucht, Germany), 2 mM L-glutamine, 100 U/mL potassium penicillin and 100 U/mL streptomycin sulfate (all from Sigma-Aldrich).

### 4.2. PBMC Stimulation

Human PBMCs were stimulated with immobile purified anti-human-CD3 (OKT3), soluble anti-human-CD28 (2 µg/mL; CD28.2; both from BioLegend, San Diego, CA, USA) or 0.35 µM sodium pyrithione (Sigma-Aldrich). For fast zinc signals, anti-CD3 was coated to beads, whereas, for activation for 24–72 h, anti-CD3 was coated to a 24-well culture plate. Anti-CD3 was coated to Dynabeads™ Pan Mouse IgG (Invitrogen by Thermo Fisher Scientific, Eugene, OR, USA) according to the manufacturer’s protocol. The beads were washed in isolation buffer (PBS + 0.1% BSA + 2 mM EDTA) and incubated for 30 min with 1 µg anti-CD3 per 1 × 10^7^ beads at 2–8 °C. Afterwards, beads were washed and stored in isolation buffer. Before usage, coated beads were resuspended in a culture medium. When anti-CD3 was coated to a 24-well culture plate, the plate was first coated with 2.6 µg/mL goat anti-mouse IgG (Jackson ImmunoResearch, Ely, UK) over night at 4 °C. Then, wells were washed with balanced salt solution (BSS) and PBS, then incubated with 2 µg anti-CD3 in 0.5 mL PBS + 0.1% BSA at 37 °C for 2 h. After washing with PBS, 1 mL with 1 × 10^6^ PBMCs were added and further stimulated with 2 µg/mL soluble anti-CD28. Cells were incubated for 24–72 h at 37 °C.

### 4.3. Labile Zinc and Calcium Measurements

To detect intracellular zinc and calcium, respectively, 10 min after stimulation, 1 × 10^6^ PBMC/mL were stained in PBS with 1 µM FluoZin-3 AM or Fluo-4 AM for 30 min (both from Invitrogen by Thermo Fisher Scientific, Eugene, OR, USA). Cells were washed and resuspended in a medium with 1 × 10^6^ PBMC/mL. Additionally, 1 × 10^6^ cells were stimulated with anti-CD28 (2 µg/mL) and/or anti-CD3 beads (1 × 10^6^ beads), while a portion of each sample was used to induce a minimal (F_min_) and maximal fluorescence (F_max_). In calcium measurements, 20 mM EDTA (F_min_; Sigma-Aldrich) and 2 µM A23187 (F_max_; Tocris, bio-techne, Minneapolis, MN, USA) were used, while in zinc measurements, 50 µM N,N,N,N-tetrakis (2-pyridylmethyl)-ethylenediamine (TPEN; F_min_; Sigma-Aldrich) and 100 µM Zinc sulfate and 5 µM sodium pyrithione (F_max_; both from Sigma-Aldrich) were used. Stimulated cells were incubated for 10 min at 37 °C in a water bath and, subsequently, the fluorescent intensity was measured using FACSCalibur™ Flow Cytometer (BD Biosciences, San Jose, CA, USA). The mean fluorescent intensity of gated lymphocytes was analyzed with FlowJo™ Software version 10.8.1 (BD Biosciences). Intracellular zinc and calcium concentrations were calculated using the following equation [55] with a dissociation constant (*K_D_*) for FluoZin-3 AM (zinc) of 8.9 nM [56] and for Fluo-4 (calcium) of 335 nM.
(1)$$\left[{Zn}^{2+}\right]\;or\;[{Ca}^{2+}]=K_{D}\times \frac{\left({F-F}_{min}\right)}{\left(F_{max}-F\right)}$$

For labile zinc and calcium measurements in cells stimulated for 24–72 h, cells were stained with FluoZin-3 AM or Fluo-4 AM, as described before. Afterwards, cells were washed and resuspended in PBS. A portion of each sample was used to measure F_min_ and F_max_, as described above, and was incubated for 10 min at 37 °C in a water bath. Stained cells were subsequently measured by flow cytometry, and intracellular zinc and calcium concentrations were calculated.

### 4.4. Kinetic Zinc and Calicum Measurements

To detect immediate zinc and calcium signaling kinetics using flow cytometry, 1 × 10^6^ PBMC/mL were stained with 1 µM FluoZin-3 AM or Fluo-4 AM, as described in Chapter 4.3. After washing, cells were adjusted to 2 × 10^6^ PBMCs in 1 mL medium. Subsequently, a baseline fluorescent signal was measured for 55 s at the flow cytometer. Then, anti-CD3 beads (2 × 10^6^) were quickly added to the cell suspension without stopping the measurement, and the measurement was then continued for a total of at least 250 s. Analysis was performed with FlowJo™ Software. First, lymphocytes were gated, including lymphocytes with increased side scatter due to binding of anti-CD3 beads. Within this gate, lymphocytes bound to anti-CD3 beads were distinguished from lymphocytes not bound to anti-CD3 beads via autofluorescence of anti-CD3 beads over 670 nm. Then, the kinetic of the mean fluorescent signal of FluoZin-3 or Fluo-4 was analyzed. For this, the mean fluorescent signal 200–250 s after measurement start was quantified and compared between lymphocytes bound to anti-CD3 beads (CD3_B_^+^) and lymphocytes not bound to anti-CD3 beads (CD3_B_^−^).

### 4.5. ELISA

For the quantification of IFN-γ, IL-10 and IL-17, supernatants were harvested after 72 h of incubation and stored at −20 °C. Samples were diluted and human IFN-γ, IL-10 (both BD Biosciences, San Diego, CA, USA) and IL-17 ELISA (R&D Systems, Minneapolis, MN, USA) were performed according to the manufacturer’s protocol. The detection limit was 4.7 pg/mL (IFN-γ), 7.8 pg/mL (IL-10) and 15.6 pg/mL (IL-17). Values below the detection limit were substituted with a value just below the detection limit. Samples were determined in duplicate, and the ELISA was measured using the Spark^®^ multimode microplate reader (Tecan, Männedorf, Switzerland).

### 4.6. Statistical Analysis

Data were statistically analyzed with GraphPad Prism (version 8.0.1). All data were tested for outliers (which were removed accordingly) before being normally distributed. Accordingly, parametric tests were selected dependent on the experimental design. The corresponding tests and the number of samples are indicated in the figure legends. Data are presented as mean + SEM. When the mean of each column was compared with the mean of every other column, letters were used to indicate significant differences. Significantly different results (*p* < 0.05) have no common identification letter. For all other tests, significances are indicated as * *p* < 0.05; ** *p* < 0.01; *** *p* < 0.001.

## Figures and Tables

**Figure 1 ijms-25-04302-f001:**
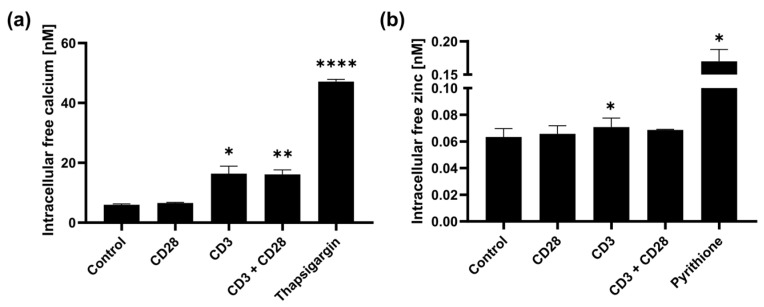
PBMCs were stained with (**a**) Fluo-4 to detect intracellular free calcium or (**b**) with FluoZin-3 to detect intracellular free zinc. Then, cells were stimulated with uncoated beads (control), soluble anti-CD28- and/or anti-CD3 coated beads and with thapsigargin or pyrithione as positive control. Signals were measured 10 min after stimulation with flow cytometry, and subsequently calcium and zinc concentrations were calculated for gated lymphocytes. Data are presented as mean + SEM with *n* = 4 (**a**) and *n* = 5 (**b**) experiments. Statistical significance to the control was determined by one-way ANOVA with Dunnett’s multiple comparisons test (* *p* < 0.05; ** *p* < 0.01; **** *p* < 0.0001).

**Figure 2 ijms-25-04302-f002:**
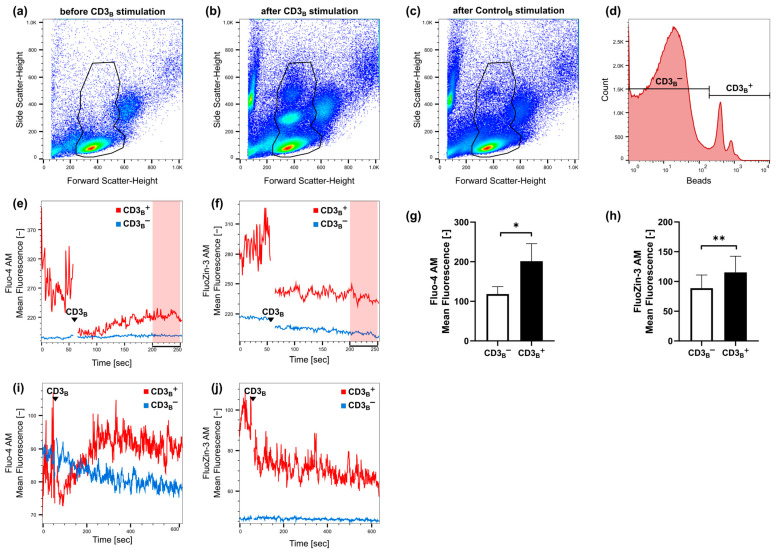
Zinc and calcium signaling kinetics after CD3 stimulation were measured. For this, PBMCs were stained with Fluo-4 (calcium) or with FluoZin-3 (zinc) and a baseline fluorescent signal was measured for 55 s with flow cytometry. Then, cells were stimulated with anti-CD3 beads (CD3_B_), while fast signals were immediately measured for at least 250 s. (**a**) An exemplary forward and side scatter density plot before stimulation with CD3_B_ is shown. The colors from red to yellow, green and blue indicate the decreasing density of events, with red representing the highest density. (**b**) After stimulation with CD3_B_, two additional populations are seen in the forward and side scatter density plot showing lymphocyte-bead complexes. Lymphocytes and lymphocyte-bead complexes were gated. (**c**) As a control, PBMCs were stimulated with uncoated beads (Control_B_). (**d**) Then, lymphocytes bound to anti-CD3 beads (CD3_B_^+^) were distinguished from lymphocytes not bound to anti-CD3 beads (CD3_B_^−^) due to an autofluorescence of the beads over 670 nm. (**e**) Exemplary calcium and (**f**) zinc kinetics are shown for CD3_B_^+^ and CD3_B_^−^. The mean fluorescent signal of (**g**) Fluo-4 or (**h**) FluoZin-3 200–250 s after stimulation with anti-CD3 beads was compared between CD3_B_^+^ and CD3_B_^−^. (**i**) Exemplary calcium and (**j**) zinc kinetics for 600 s (10 min) are shown of *n* = 2 experiments. Data are presented as exemplary experiments (**a**–**f**,**i**,**j**) or as mean + SEM with *n* = 7 (**g**) and *n* = 8 (**h**) experiments. Statistical significance to the control was determined by paired t-test (* *p* < 0.05; ** *p* < 0.01).

**Figure 3 ijms-25-04302-f003:**
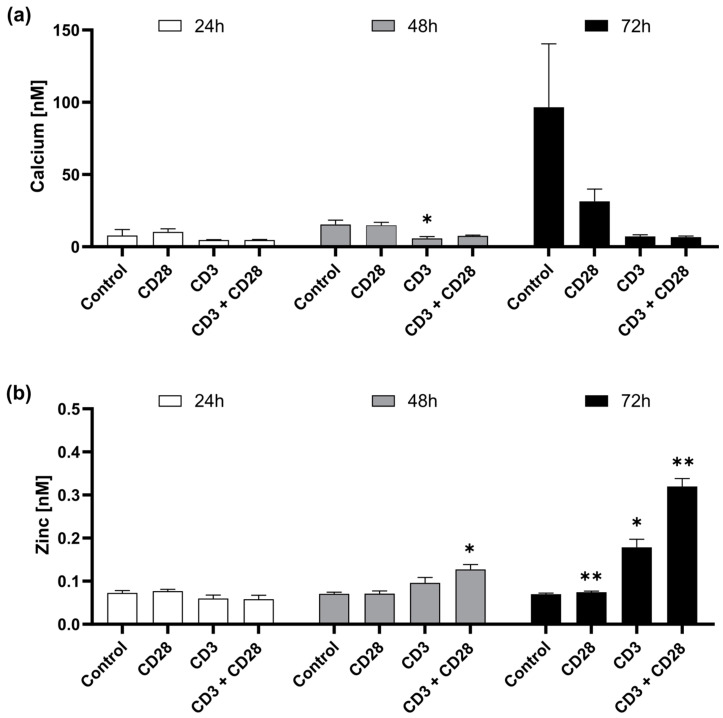
PBMCs were stimulated with immobile anti-CD3 and/or soluble anti-CD28 and incubated for 24, 48 or 72 h. After the respective incubation times, intracellular free (**a**) calcium and (**b**) zinc were measured by Fluo-4 and FluoZin-3, respectively, with flow cytometry. Data are presented as mean + SEM with *n* = 4–5 (**a**) and *n* = 4 (**b**) experiments. Statistical significance to the control was determined by two-way ANOVA with Dunnett’s multiple comparisons test (* *p* < 0.05; ** *p* < 0.01).

**Figure 4 ijms-25-04302-f004:**
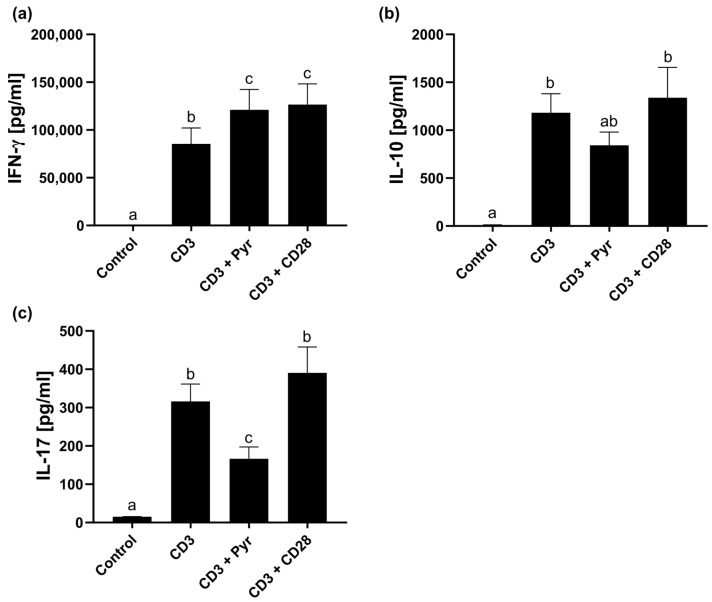
PBMCs were stimulated with immobile anti-CD3 and/or soluble anti-CD28 and 0.35 µM pyrithione (Pyr). After 72 h, IFN-γ (**a**), IL-10 (**b**) and IL-17 (**c**) were measured in the supernatant by ELISA. Data are presented as mean + SEM with *n* = 10 (**a**) and *n* = 9 (**b**,**c**) experiments. Statistical significance was determined by one-way ANOVA with Tukey’s multiple comparisons test (**a**,**c**) or a Friedman test with Dunn’s multiple comparisons test (**b**). Significantly different results (*p* < 0.05) have no common identification letter.

## Data Availability

The data presented in this study are available on request from the corresponding author. The data are not publicly available due to data repository not already active at the faculty.

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
