# Peer review of "Zinc Ionophore Pyrithione Mimics CD28 Costimulatory Signal in CD3 Activated T Cells"

_ijms, 2024, doi:10.3390/ijms25084302_

Round 1
Reviewer 1 Report
Comments and Suggestions for Authors
In the present manuscript, Jacobs et al. investigated fast and long-term changes in intracellular zinc and calcium in TCR/CD3 and CD28-stimulated PBMC.
The authors describe following findings:
(i) Fast zinc fluxes are induced via CD3-stimulation,
(ii) CD3 and CD28 signaling induce a homeostatic zinc signal,
(iii) The zinc ionophore pyrithione mimics the effect of the CD28 costimulatory signal in CD3 activated T cells regarding IFN-gamma (TH1) production and secretion.
I have some comments and questions to this manuscript.
(i) The data in Figure 1(b), showing that CD3 stimulation over 10 min induces an significant increase in intracellular free/labile zinc are not really convincing. It would help to improve the manuscript, if the authors could present time point kinetic data (e.g. 2, 5, 10, 15 min) of the zinc flux after CD3 stimulation.
(ii) Figure 2 is relatively complicated to understand. The figure would be improved, if the authors would show additionally:
- Lymphocytes (PBMC) without beads (Side Scatter vs. Forward Scatter),
- FluoZin-3 vs. Time also for the zinc measurement.
(iii) Section 2.2 with Figure 3 is not clearly written.
E.g. the sentence “However, CD3 and CD3-CD28 stimulation resulted in a decreasing trend in intracellular free zinc” is not clear to me. I see a clear increase after 72 h. Please explain.
(iv) The experiment in Section 2.3 (Figure 4) is very easy and the data convincing. But it would be nice to know, whether or not pyrithione mimics also the expression of other cytokines like IL-2 (TH1), IL-5 (TH2), IL-17 (TH17).
Author Response
POINT-BY-POINT-Reply
We thank the reviewers and the editor for the overall positive feedback. We changed the manuscript in all points possible within the revision time of 10 days and hope that the revised manuscript is acceptable for publication.
During editing, we noticed that the SD (standard deviation) was inadvertently shown in Fig.1a and Fig.3b, but SEM (standard error of mean) was indicated in the figure legend. Now we corrected the picture to SEM. We apologize for this inconvenience, but in fact the data were presented before less strong than now.
Reviewer 1
- COMMENT: The data in Figure 1(b), showing that CD3 stimulation over 10 min induces an significant increase in intracellular free/labile zinc are not really convincing. It would help to improve the manuscript, if the authors could present time point kinetic data (e.g. 2, 5, 10, 15 min) of the zinc flux after CD3 stimulation.
RESPONSE: We completely agree with the reviewer that a closer investigation of the zinc signaling kinetics is necessary. For this reason we performed kinetic measurements with flow cytometry which measured signals continuously from 0 – 600 sec (10 min) (Fig. 2). We specifically analyzed the short time point 200 - 250 sec (3.3 – 4.2 min) in Fig. 2 g+h and added kinetic signals up to 10 min (Fig. 2 i+j). Signals were stable for 10 min so we did not investigate longer time periods.
- COMMENT: Figure 2 is relatively complicated to understand. The figure would be improved, if the authors would show additionally:
- Lymphocytes (PBMC) without beads (Side Scatter vs. Forward Scatter),
- FluoZin-3 vs. Time also for the zinc measurement.
RESPONSE: We thank the reviewer to indicate that this figure is complicated to understand. We added the requested figures and additionally showed a stimulation with uncoated beads (Fig.2c). We hope that this clarifies the figure and transparently presents our analysis.
- COMMENT: Section 2.2 with Figure 3 is not clearly written.
E.g. the sentence “However, CD3 and CD3-CD28 stimulation resulted in a decreasing trend in intracellular free zinc” is not clear to me. I see a clear increase after 72 h. Please explain.
RESPONSE: We thank the reviewer for bringing this mistake to our attention. The reviewer is right and the sentence has been corrected.
- COMMENT: The experiment in Section 2.3 (Figure 4) is very easy and the data convincing. But it would be nice to know, whether or not pyrithione mimics also the expression of other cytokines like IL-2 (TH1), IL-5 (TH2), IL-17 (TH17).
RESPONSE: We thank the reviewer for this complementary input. We have extended the experiment with IL-10 (TH2) and IL-17 (TH17) ELISA and thank you for extending the informative value.
Reviewer 2 Report
Comments and Suggestions for Authors
The authors have performed good research on effects of zinc to improve T cell immunity. There is need some minor corrections as highlighted in the file.
The authors have used t-test for significance of results, why not used ANOVA for significance.

Comments on the Quality of English LanguageMinor corrections needed
Author Response
POINT-BY-POINT-Reply
We thank the reviewers and the editor for the overall positive feedback. We changed the manuscript in all points possible within the revision time of 10 days and hope that the revised manuscript is acceptable for publication.
During editing, we noticed that the SD (standard deviation) was inadvertently shown in Fig.1a and Fig.3b, but SEM (standard error of mean) was indicated in the figure legend. Now we corrected the picture to SEM. We apologize for this inconvenience, but in fact the data were presented before less strong than now.
Reviewer 2
- COMMENT: The authors have performed good research on effects of zinc to improve T cell immunity. There is need some minor corrections as highlighted in the file.
RESPONSE: We thank the reviewer for the overall positive feedback and for helping to improve the English language. We corrected all the addressed errors.
- COMMENT: The authors have used t-test for significance of results, why not used ANOVA for significance.
RESPONSE: We thank the reviewer for checking the statistical tests. For multiple comparisons of the mean between three or more groups we used the one-way ANOVA (Fig. 1 and 4) or two-way ANOVA test (Fig. 3). We used the t-test only for comparing the means of two groups (Fig. 2).
Round 2
Reviewer 1 Report
Comments and Suggestions for Authors
The authors answered on all comments.
The manuscript is now suitable for publication.